# Humans and the Olfactory Environment: A Case of Gene-Culture Coevolution?

**Peter Frost** 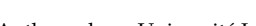

Anthropology, Université Laval, Quebec, QC G1V 0A6, Canada; peter_frost61z@telus.net

**Abstract:** As hunter-gatherers, humans used their sense of smell to identify plants and animals, to find their way within a foraging area, or to distinguish each other by gender, age, kinship, or social dominance. Because women gathered while men hunted, the sexes evolved different sensitivities to plant and animal odors. They also ended up emitting different odors. Male odors served to intimidate rival males or assert dominance. With the rise of farming and sedentism, humans no longer needed their sense of smell to find elusive food sources or to orient themselves within a large area. Odors now came from a narrower range of plants and animals. Meanwhile, body odor was removed through bathing to facilitate interactions in enclosed spaces. This new phenotype became the template for the evolution of a new genotype: less sensitivity to odors of wild plants and animals, lower emissions of male odors, and a more negative response to them. Further change came with the development of fragrances to reodorize the body and the home. This new olfactory environment coevolved with the ability to represent odors in the mind, notably for storage in memory, for vicarious re-experiencing, or for sharing with other people through speech and writing.

**Keywords:** gene-culture coevolution; odor; olfaction; perfume; recent evolution; sex differences

## 1. Introduction

The sense of smell is underappreciated. Though less crucial than sight or hearing, it tells us about what we neither see nor hear. It also enriches sight and hearing with biochemical data on objects of interest. Finally, by producing disgust or pleasure, it helps us decide whether such objects should be avoided or approached.

The act of smelling can indeed be pleasurable. This psychological reward has led us to remake our olfactory environment in ever more appealing ways, just as we have remade our visual environment with pure bright colors that are rare in nature. The process began with the human body, which we learned to deodorize by bathing in water and then reodorize by perfuming with aromatic compounds. Reodorization would eventually spread to the entire home, with the result that we now spend much, if not most, of our time in an olfactory environment of our making.

The change has not been all one-way. By remaking our olfactory environment, we have ended up remaking ourselves. The new olfactory environment has made some human groups more sensitive to certain odors and less sensitive to others, while reducing the capacity to emit body odors and enhancing the ability to experience certain odors vicariously upon hearing or seeing them mentioned by name. We have thus coevolved with our olfactory environment.

That coevolution raises several questions:

- How do odor emission and odor detection differ among humans? How do they differ between men and women, between age groups, and between populations?
- To what degree are these differences culturally influenced, and to what degree are they genetically influenced?
- What is gene-culture coevolution? How can a cultural difference between two groups facilitate the evolution of a genetic difference?

- What are the trajectories of gene-culture coevolution within our species? How has the sense of smell coevolved with the olfactory environment? How has the ability to represent odors in the mind coevolved with the olfactory environment?

Those questions have already aroused the interest of some researchers, notably Jan Havlíček and S. Craig Roberts. To a lesser degree, they have also interested researchers who work from other theoretical perspectives in psychology and social anthropology, such as Joël Candau and Andreas Keller. In general, there is a growing awareness that the relationship between humans and their environment is not one-way. This awareness has begun to influence the focus of research in a wide range of fields, including research on human olfaction.

To address the above questions, it is important to understand certain key concepts, which are summarized in Table 1.

To understand this coevolution, we should begin where it began, with ourselves. Our body odors provide information on personal identity: gender, age, degree of kinship, ethnicity, and even personality. Because such information matters more to some of us than to others, sensitivity to body odors has come to differ from one individual to another, as has the capacity to emit them. Odor emission and detection thus form a signaling system for the exchange of information among men, among women, between men and women, or between age classes (see Table 2).

**Table 1.** Key Concepts.

| | |
|---|---|
| Farming | A means of subsistence that different human groups began to adopt 10,000 years ago. It involved the domestication of certain plant and animal species, with the result that humans encountered a narrower range of odors. It also resulted in sedentism, population growth, and increasing social complexity. |
| Gene-culture coevolution | The reciprocal evolution of genes and culture. Humans adapt not only to their natural environment but also to their cultural environment. There thus develops a feedback loop: humans create a growing proportion of their environment, and this human-made environment increasingly modifies the human genome through natural selection. |
| Hunting and gathering | The means of subsistence of all humans until about 10,000 years ago. In general, men hunted wild animals, and women gathered wild plants (seeds, berries, roots, tubers, etc.). |
| Odor detection | See Olfaction |
| Odor emission | The release of volatile organic compounds (VOCs) into the air. VOCs include volatile androgens in the case of men (e.g., androstenone, androstadienone, androstenol) and volatile estrogens in the case of women (e.g., estratetraenol). Most VOCs are organic acids whose function remains largely unknown. |

**Table 1.** *Cont.*

| | |
|---|---|
| Odor memory | The mental representation of an odor, its storage in memory, its retrieval at a later date, and the re-experience of it. |
| Olfaction | The sense of smell. It has three components: (1) sensitivity (minimum number of molecules needed, per volume of air, to identify the presence of a volatile compound); (2) pleasantness or unpleasantness (degree to which a volatile compound is evaluated positively or negatively); and (3) discrimination (ability to distinguish between differently structured volatile compounds). |
| Perfume | A human-prepared mixture of volatile compounds that serves to give the body and the home a pleasant smell. It includes not only solvent-based fragrances but also incenses, scented balms, and aromatic bath oils. |
| Recent evolution | Evolution of human groups during the time of recorded history. |

Such information is often exchanged within a reproductive context. It may serve to avoid inbreeding, to assess potential mates, or to ward off sexual rivals [1]. In other cases, as we will see further, the context is less directly reproductive and may include nonhuman emitters, such as plants and animals that may be identified by smell as food sources. The signaling system thus extends beyond human bodies and into the "extended phenotype" of culture: initially, wild plants and animals that supplied food, medicine, and material for clothing or shelter; later, crops and livestock for the same purposes, as well as labor for certain tasks; and, finally, indoor spaces, which have been progressively deodorized and reodorized. Over time, this extended phenotype has taken up a growing proportion of our olfactory environment.

**Table 2.** Odor Emission/Detection by Gender, Age, and Ethnicity.

| Odor Emitter | Function/Effect of Body Odor |
|---|---|
| Men | Male odor signals male presence ($5\alpha$-androstenone) [2]. |
| | Male odor decreases cooperativeness among men for roles that can provoke aggression and competition (androstadienone) [3]. |
| | Male odor increases cooperativeness among men for submissive roles [4]. |
| | Male odor increases gaze avoidance by socially anxious men [5]. |
| | Male odor increases perceived dominance of male faces when viewed by socially anxious men [6]. |
| | Male odor increases women's preference for masculinized faces [7]. |
| | Both sexes respond with anxiety to the odor of men who have been boxing and not simply exercising [8]. |
| | Men respond with anxiety (higher skin conductance) to the odor of men who have been playing badminton and not simply running. Response correlates with predisposition to social anxiety [9]. |
| | Women in the fertile phase of their menstrual cycle prefer the odor of men who score high on social dominance [10]. |

**Table 2.** *Cont.*

| Odor Emitter | Function/Effect of Body Odor |
|---|---|
| Women | Female odor increases cooperation by men with other men (estratetraenol) [11]. |
| | The odor of women's tears makes female faces less sexually appealing to men, while decreasing testosterone levels, sexual arousal, and neural activity in brain areas associated with sexual arousal. There are no such responses to the odor of saline solution [12]. |
| | By smell alone, women can distinguish between underarm pads worn by women who have watched a horror film and underarm pads worn by women who have watched a neutral film [13]. |
| Mother's breast | By smell alone, infants can recognize their mother's breast, apparently through rapid learning [14]. |
| Older age groups | By smell alone, young men and women can identify elderly individuals, whose odor seems less intense and less unpleasant than that of younger individuals [15]. |
| Close kin | By smell alone, fathers, grandmothers, and aunts can identify garments previously worn by their neonatal relatives. They can accurately distinguish the smell of a stranger from that of a same-sex/same-age sibling who has been separated from them for 1 to 30 months [16]. |
| Ethnic groups | Sub-Saharan Africans emit the largest quantity of body odorants, followed by Europeans and then East Asians [17,18]. |
| **Odor Detector** | **Differences between Groups** |
| Women vs. men | Women have a keener sense of smell, perhaps due to the sexual division of labor among hunter-gatherers: women gathered plant foods while men hunted game animals. Women thus discriminated between edible and inedible items at close range, whereas men relied more on sight to pursue game animals. Men used smell mostly to detect the scent trail of smaller animals [19–23]. |
| Younger vs. older | Olfactory acuity declines with age [19–21]. |
| Europeans vs. Africans | Androstadienone, a volatile androgen, is stronger-smelling to African Americans than to European Americans. The latter have alleles for different degrees of sensitivity to the aromatic compound methanethiol (asparagus odor), whereas African Americans uniformly have the allele for the least sensitivity [20,24]. |
| East Asians vs. Europeans | Asian Americans are more sensitive than European Americans to several related compounds: nonyl aldehyde, decyl aldehyde, and undecanal. Japanese are better at describing "Japanese" odors and Germans at describing "European" odors, with the two groups clearly differing in ratings of pleasantness [20,25,26]. |
| Europeans vs. Amerindians | Tsimane' from the Bolivian rainforest can detect n-butanol at lower concentrations than Germans, and 25% of them are more sensitive than any German [27]. |

## 2. Odor Emission and Detection: Culturally or Genetically Influenced?

How much of this signaling system is learned? How much innate? There is no easy answer, partly because genetic and cultural influences interact with each other, and partly because a genetic influence can vary from one stage of mental processing to another, from one odor to another, and from one individual to another.

First, even when odor detection is under tight genetic control, it still takes place within a broader cultural context that influences how the odor is evaluated. Evaluation may be positive in one context but negative in another [28].

Second, culture can change the genetic makeup of a population. Some people are a better "fit" for their culture because they can readily learn its rules and exploit the opportunities it creates. They thus enjoy reproductive success. As their behavioral traits spread through the gene pool, the culture will meet with less resistance from the average individual and will shift further in the same direction as the behavioral shift. Cultural evolution is therefore difficult to separate from genetic evolution. The two support each other, forming, in fact, a single trajectory of gene-culture coevolution.

Third, a genetic influence can vary from one stage of mental processing to another. In some cases, as with the odor of a mother's breast or close kin, we learn to recognize a specific smell, but that learned information is fed into an innate algorithm that allows recognition of only one smell or a limited number [14,16]. The algorithm "expects" information from the environment. Once that information has been supplied, it loses superfluous neural units and becomes fully hardwired [29]. For instance, prenatal exposure to an odor can produce permanent recognition of that odor [30]. In other cases, we recognize a specific smell without any learning; the algorithm is fully hardwired from the outset [31]. The rule seems to be that natural selection tends to hardwire any mental task that is both frequent and predictable. In such cases, the advantages of learning do not offset the disadvantages, notably the time required for learning and the mistakes that will inevitably occur during that time.

Fourth, a genetic influence can vary from one odor to another [20,32–34]. A twin study found that sensitivity to two odors is 78% and 73% heritable [32]. In the case of another odor, a single gene explains over 96% of the observed variation in sensitivity [28]. Yet there may be little or no heritable variation in sensitivity to other odors [35].

Finally, in addition to varying by sex and age, a genetic influence can vary from one individual to another [33,36]. For some people, androstenone smells like sweat or urine. For others, it has a pleasant sweet or floral smell. For still others, it is odorless [33]. An American study found that 63% of odorant receptors are functionally polymorphic and that, on average, over 30% differ between any two individuals. African Americans were found to be the most genetically diverse group in this respect, although much of that genetic diversity does not translate into functional differences with real effects [37].

## 3. Gene-Culture Coevolution and Population Differences

Can a genetic influence vary from one human population to another? Joël Candau, known for his work on the "anthropology of odors," has pointed to population differences as an argument against genetic determination. Citing the example of the durian fruit, which is popular throughout Southeast Asia but smells like putrefaction to Westerners, he concludes that olfactory preferences are strongly shaped by experience, culture, and context [38] (p. 52), [39]. Certainly, culture does play a role, as shown by the finding that Chinese and Indians are not as sensitive to certain odors in Germany as they are in their home countries [26].

A cultural cause does not, however, preclude a genetic one. The two can coexist, with the former paving the way for the latter. Again, this is gene-culture coevolution. When humans enter a new territory, they must adapt not only to a new natural environment but also to the cultural environment they are creating, including the odors of new kinds of food, shelter, and clothing, as well as certain practices, such as bathing, that reduce body odor. At first, they adapt within the limits of their genotype by doing the best they can with what they have, i.e., by pushing some abilities beyond their normal limit and by behaving in uncomfortable but doable ways. In short, they force-fit their genotype into the phenotype they themselves have created. The phenotype will then act as a template for selection: the better you fit into it, the better you will do in life, and the more progeny you will have. Over successive generations, the mean genotype will correspond more and more to the human-created phenotype. Each culture thus has its own evolutionary trajectory.

We know that olfactory evolution has followed different trajectories in different human populations. The main evolutionary split seems to exist between Eurasians and sub-Saharan

Africans. The latter emit the highest levels of volatile organic compounds (VOCs) from their bodies, followed by Europeans and then East Asians [17]. These differences are mirrored by differences in apocrine gland production [18]. In the case of male bodies, the VOCs include volatile androgens such as androstenone and androstenol, although most are organic acids whose function remains largely unknown [17]. In addition to having the highest emissions of VOCs, sub-Saharan Africans also have the largest number of functional olfactory receptors [40] and the highest diversity of alleles for olfactory receptors [37]. Furthermore, they have a structurally different receptor for volatile androgens, which may explain their higher sensitivity to androstadienone, a likely human pheromone [20], [33] (supp. Methods), [41]. Conversely, Eurasians are much more likely to have alleles for reduced sensitivity to volatile androgens [42]. It may be that sub-Saharan Africans have retained an emission/detection profile that all humans once had: high levels of emitted VOCs and high sensitivity to their presence in the air.

The African-Eurasian split was followed by later divergences within each of these two major groups. For example, a cross-cultural study found that Japanese participants tended to rate "Japanese" odors as more intense, while German participants tended to rate "European" ones as more intense. These tendencies nonetheless had interesting exceptions. Three Japanese odors (dried fish, Japanese tea, soybeans) seemed more intense to the Germans, whereas one European odor (church incense) seemed more intense to the Japanese [25]. Such exceptions might exist because genetic evolution has lagged behind recent cultural changes in one population or the other.

In addition to odor sensitivity, populations may differ in how positively or negatively they perceive an odor or how easily they can tell it apart from a related one [25]. Detection thus has three components: sensitivity (minimum number of molecules needed, per volume of air, to identify the presence of a volatile compound); pleasantness or unpleasantness (the degree to which a volatile compound is evaluated positively or negatively); and discrimination (ability to distinguish between differently structured volatile compounds).

## 4. Coevolution between the Olfactory Environment and the Sense of Smell

In this case, as in others, gene-culture coevolution explains much of recent human evolution. More than any other animal, we make the world we live in, and it is often the human-made portion of our environment that decides who lives to reproduce. By remaking our surroundings, we have ended up directing our evolution [43–45].

With respect to the olfactory environment, gene-culture coevolution can be divided into three historical stages: hunting and gathering, farming, and reodorization of bodies and homes.

### 4.1. Stage 1: Hunting and Gathering

Early humans did not live in a world of their making. They lived in a natural environment where their sense of smell helped them hunt and gather within a large foraging area. For instance, the odor of a lake or a forest could mark their way to a destination and the way back. Among present-day hunter-gatherers, such as the Umeda of New Guinea, smell is as good as sight for spatial orientation. It is often better for some tasks, such as detecting a distant campfire [46] (p. 98).

Smell also enables hunter-gatherers to identify people by age, gender, and social dominance. Among the Suya of central Brazil, men are expected to have a bland smell, which is considered to be a condition for living in society. A strong smell is normal for some Suya: for children, because they have not yet been socialized; for the elderly, because they are no longer subject to social restrictions; and for women, because they are seen as living outside society. Tribal leaders, too, are said to have a strong smell, which indicates their power not only to rise above the social order but also to inflict social disorder [47] (pp. 106–120, 202–203). Finally, smell may indicate membership in a tribal group. According to the hunter-gatherers of northwest Amazonia, each group has its own odor and can mark its territory with an "odor-thread" [48] (pp. 125–126).

The importance of smell is shown by the vocabulary of hunter-gatherers, such as the Jahai of peninsular Malaysia. Their language has a dozen stative verbs for specific odors: "to be fragrant," "to be musty," "to have a stinging smell," "to have a urine-like smell," etc. "These verbs are common parlance, known and used by all. Everyday conversation is peppered with them and they are not limited to religious, mythical, or otherwise specialist genres" [23] (pp. 356–357), [49] (pp. 24–25), [50].

The literature on hunter-gatherers occasionally refers to adults removing body odors through bathing, which is usually confined to special occasions. Among hunter-gatherers of the northwest coast of North America, such occasions include preparing for a bear hunt, searching for a guardian spirit, ending a period of mourning, or being initiated into a secret society [51] (pp. 12–13), [52] (pp. 892, 899). Sweat lodges have been similarly used by North American indigenous peoples to cleanse themselves for healing or ceremonies [53]. In Amazonia, bathing is a pre-hunting ritual that men perform to hide their odor from their prey [54] (p. 19). In other groups, such as the Hadza of Tanzania, bathing is rare because suitable water is scarce [55] (p. 428). In central Africa, frequency of bathing is seen as a distinction between the Efe hunter-gatherers and the neighboring Lese, a farming people: "Lese women noted that whereas Lese men wash their hands and bodies frequently, sometimes once a day, the Efe may bathe only once a week and do not use soap" [56] (p. 79). Whenever non-ritual adult bathing has been reported among hunter-gatherers, there is evidence of outside influence, such as the use of soap or shampoo [57] (p. 142).

In sum, body odor is considered most problematic during hunting when it can alert game animals to the presence of humans. This is the stated reason why women should not hunt: the scent of menstrual blood is said to drive animals away [58]. Body odor is also problematic during encounters with the spirit world. The least problematic encounters happen in the course of normal social relations, which are not seen as a reason for deodorizing the body.

### 4.2. Stage 2: Farming

Beginning some 10,000 years ago, hunting and gathering gave way to farming, which, in turn, led to sedentism, population growth, and increasing social complexity [43–45,59]. People no longer needed their sense of smell to find elusive food sources or to orient themselves within a large area. Their senses of sight or hearing were usually enough. Moreover, with the replacement of wild foods with crops and livestock, they now encountered a narrower range of plant and animal odors.

This new means of subsistence would follow different trajectories in different regions, with the main split being between sub-Saharan Africa and Eurasia.

### 4.2.1. Sub-Saharan Africa

Farming was largely women's work in sub-Saharan Africa [60]. Being possible year-round, it enabled a mother to feed herself and her offspring with a minimum of male assistance. The low cost of wife maintenance incentivized men to seek as many women as possible, with the result that 20–50% of all marriages were polygynous in traditional sub-Saharan farming societies [61] (p. 51), [62–65]. In his survey of the subject, the anthropologist Jack Goody concluded that "hoe agriculture, female farming and polygyny are clearly associated in a general way." He added that high polygyny rates persist nonetheless in some sub-Saharan societies where men have become substantially involved in farming [62] (p. 185).

A high polygyny rate has consequences for all men, whether married or single. A field study in Tanzania has shown behavioral and even physiological similarities between married men in a polygynous society and single men in a monogamous society: "In hunter-gatherer societies, such as the monogamous Hadza of Tanzania (Africa), men invest more in offspring than in small-scale pastoralist societies, such as the polygynous Datoga of Tanzania. Polygyny and between-group aggression redirect men's efforts from childcare toward investment in male-male relationships and the pursuit of additional mates. When

men participate in childcare, their testosterone (T) level decreases. [ . . . ] [A]mong the monogamous, high paternally investing Hadza, T levels were lower for fathers than for non-fathers. This effect was not observed among the polygynous, low paternally investing Datoga" [66] (pp. 1–2). Wherever polygyny is prevalent, so is male rivalry for access to women, as indicated by testosterone levels. There is consequently a high level of male aggression: "Datoga males reported greater aggression than Hadza men—a finding in line with previous reports. [ . . . ] There is a negative attitude toward aggression among the Hadza but not among the Datoga. In situations of potential aggression, the Hadza prefer to leave. In contrast, aggression is an instrument of social control—both within the family and in outgroup relations in Datoga society. Datoga men are trained to compete with each other and to act aggressively in particular circumstances" [66] (p. 6).

Thus, as hunting and gathering gave way to farming in sub-Saharan Africa, there were successive increases in the polygyny rate, in male rivalry, and in male aggression. Acts of aggression were not repressed by the State because state formation came relatively late, in no small part because of the disruption due to neighboring communities raiding each other for women. Raiding parties were typically young men who saw warfare as their only way out of celibacy [67] (pp. 50–51). A cross-cultural survey has shown that "polygyny is associated with warfare for plunder and/or female captives" [68] (p. 882). Raided communities were disrupted not only by the loss of young women but also by the deaths of younger men and older women, who were less likely to be spared [68] (p. 874). Over successive generations, a high level of male rivalry would favor certain physical adaptations to help men fight rivals or assert dominance [69] (pp. 182–183). One adaptation would include higher emissions of VOCs [17,18,42].

### 4.2.2. Eurasia

In Eurasia, as in sub-Saharan Africa, farming narrowed the range of plant and animal odors in the environment. In Eurasia, this range narrowed even further with the rise of trade and specialization. Large areas of farmland became monocultures as part of a change that began in the Middle East, especially for cereals such as barley, einkorn wheat, and emmer wheat.

Eurasia differed from sub-Saharan Africa in two other ways. First, male rivalry for access to women remained at a relatively low level. Because men contributed more to food production and provided for their families to a greater extent, only a few could afford multiple wives [62] (pp. 176–177). Second, states were formed at a much earlier date, particularly within a zone stretching from the Mediterranean, through the Middle East, and into South and East Asia. Those states sought to monopolize the use of violence, seeing the aggressive male as a threat not only to society but also to their power [70–72]. A man no longer had the right to act aggressively at will, except to defend himself and his family. Endemic violence was thus kept at a lower level in Eurasia than in sub-Saharan Africa.

Because Eurasian groups were less prone to male rivalry, they became less receptive to the presence of volatile androgens. These VOCs increase male cooperation for submissive tasks but decrease it for competitive tasks which require high levels of coordination and cooperation [3,4]. The latter tasks tended to predominate in communities across Eurasia [73], with the result that men sought to remove their body odor through regular bathing, first in rivers and later in public bathhouses [74,75] (pp. 59, 110–111), [76,77]. For both sexes, this practice helped facilitate interactions in enclosed spaces with people who were not necessarily close kin. As more people began to bathe, it was done not only to smell "better" but also to be "normal." The new normal eventually affected not only social relations but also the genetic makeup of the local population by favoring those men who naturally emitted lower levels of volatile androgens and who were more negatively disposed toward them.

In sum, the olfactory environment became more controlled in Eurasia, being characterized not only by crop and livestock monocultures but also by human monocultures—large numbers of people living in a single society under state domination. There was thus a nar-

rower range of odors from plants and animals, together with constraints on the acceptability of volatile androgens.

*4.3. Stage 3: Reodorization of Bodies and Homes*

A third stage began when people not only removed their natural odors but also added new ones by means of perfumes, incenses, scented balms, and aromatic bath oils. The new scents often served to increase sexual attractiveness, perhaps by interacting with some of the body's natural odors [78,79]. In a compendium that included descriptions of perfumes, written in India in the 6th century, the section on perfumery was sandwiched between sections on aphrodisiacs and sexual intercourse [80] (pp. 72–73). For whatever reason, the practice became widespread. By the time of the Greek historian Herodotus, in the 5th century BC, the men of Babylon were said to be "anointed over the whole of their body with perfumes" [81] (*Histories* I: 195). Around the same time, the inhabitants of the Indian city of Ayodhya were described in similar terms: "There was no one who was dirty or whose body lacked for ointments or perfumes" [80] (p. 57). As body odors became less noticeable, not only through regular bathing but also through regular perfuming, the capacity to emit them further atrophied.

By perfuming themselves, women reoriented sexual interest toward their upper body, particularly the face. This point is made by Havelock Ellis in his writings on the psychology of sex: "The odor of the body, like its beauty, in so far as it can be regarded as a possible sexual allurement, has in the course of development been transferred to the upper parts. The careful concealment of the sexual region has doubtless favored this transfer" [82]. Reorientation toward the face was further assisted by romantic kissing, a practice unknown to most hunter-gatherers [83]. As an expression of erotic desire, kissing seems to have first gained broad acceptance in the ancient civilizations of the Mediterranean, the Middle East, and South Asia [82,84,85].

In time, perfumes were used to reodorize not only the body but also the home environment. People came to control not only the temperature and lighting of their homes but also the smell. In 1843, while traveling through Hadramaut, a European observer noted that rooms were fumigated five or six times a day with frankincense [86]. Homes are still regularly fumigated with frankincense and myrrh in northeast Africa and south Arabia [87]. In the United Arab Emirates, a research team found that 86% of all households burn incense indoors at least once a week [88]. The same researchers found no consistent associations between incense use and wheezing, coughing, or shortness of breath, perhaps because the Emiri population has adapted to this practice [88].

At an early date, reodorization was extended to spaces outside the home through the planting of aromatic flowers and shrubs. The Persians were pioneers in this respect, as attested by the Greek historian Xenophon when describing a Persian garden: "Lysander was astonished at the beauty of the trees within, all planted at equal intervals, the long straight rows of waving branches, the perfect regularity, the rectangular symmetry of the whole, and the many sweet scents which hung about them as they paced the park" [89] (*Oeconomicus* 1:4.16–18). Throughout the Middle East and the Mediterranean, "pleasure gardens" were planted with aromatic shrubs, such as jasmines [90].

## 5. A Second Coevolution: The Olfactory Environment and the Ability to Represent Odors in the Mind

The new olfactory environment coevolved not only with the sense of smell but also with the ability to represent odors in the mind. Individuals had to become better at:

- representing certain odors in their minds;
- exchanging these mental representations with other people via speech and, later, writing;
- storing these representations in memory;
- recalling them at a later date in full detail.

In this case, as in others, the coevolution was at first non-genetic: people made better use of their existing abilities, in particular the ability to develop thoughts collectively. This

is a corollary of the Sapir–Whorf hypothesis: language is not merely a passive medium of thought; it is a means to express thought in a more organized, coherent, and systematic manner through the very process of transforming mental representations into words, including the effort of making one's thoughts understandable to people who think differently and who assist by offering comments, corrections, or additional information.

Thus, as language developed, so did thought, including the mental representation of odors. Once an odor was named, it became a concept that could be manipulated in speech and mind, notably through comparison with other concepts: "a named odor has more chances of being categorized and these chances increase when the naming is precise" [38] (p. 52). An odor could also be re-experienced simply upon seeing or hearing its name; this vicarious experience could then become a thing to be savored, enjoyed, and shared with other people, including those who had never known it in real life. The odor experience could now circulate within a larger space—not only the internal memory of each person's mind but also the external memory of oral and written tradition [91].

In addition to a purely cultural coevolution, was there also gene-culture coevolution? Just as natural selection improved the ability to discriminate between certain odors, did it also improve the ability to represent them in one's mind, speech, and writing? There may, for instance, have been selection for individuals who could more readily name an odor, recall it later in full detail, and, thus, re-experience it vicariously. Indeed, according to several studies, a named odor is more strongly remembered than a nameless one [92–96]. One study, however, failed to replicate this finding [97]. It has been suggested that the replication failure might have a methodological cause, particularly the use of forced choices [98]. Another possibility is that these different studies used subjects from populations with different trajectories of gene-culture coevolution.

The mental representation of an odor thus has an existence that is distinct from that of the odor itself. It can take on a life of its own. The decoupling between signifier and signified seems to have begun early in cultural evolution. In an essay on totemism, the anthropologist Claude Lévi-Strauss argued that animal totems are "good to think," just as delicacies are good to eat. Whenever a hunter-gatherer thought about a particular animal, the thought could, in itself, arouse an emotional response, either positive or negative. Certain animals thus became totems because of their positive emotional value [99] (p. 89). Similarly, when people think about a pleasant-smelling odor, they are motivated to increase its presence in their environment, thereby increasing their encounters with it. A process of positive feedback thus begins. As the odor becomes more often encountered and the pleasurable response increasingly accessible through memory, the stimulus-response sequence becomes more predictable, even automatic. The advantages of learning no longer outweigh the disadvantages. Hardwiring is now the better option.

The question here is whether some emotional responses have become more hardwired over the past 10,000 years, such as responses to fragrances used for bodies and homes, or less hardwired, such as responses to scents produced by wild plants and animals. Given the short timescale, natural selection probably altered existing hardwiring, as opposed to starting from scratch [100]. For instance, the emotional response to an odor may have atrophied in some cases while becoming stronger in others. Or the threshold for response may have been raised or lowered. Finally, there may have been selection for individuals who could more readily recall the smell of a certain desirable odor and feel the corresponding emotional response simply upon hearing or seeing its name. Selection may have thus targeted not only the response to an odor but also the mental representation of that odor and the ability to generate that mental representation.

*Coevolution of Language with the Olfactory Environment in the Middle East*

To understand the coevolution between the olfactory environment and the representation of odors in the mind, we can take the example of the Middle East. This was the region where odor experiences were first shared in speech and writing, as a result of the "first

social media revolution" [91]. A Sumerian poem from the third millennium describes the goddess Inanna making preparations for a visit from her beloved:

> O that someone would tell my mother,
> and she sprinkle cedar perfume on the floor,
> O that someone would tell my mother Ningal,
> and she sprinkle cedar perfume on the floor!
> Her dwelling, its fragrance is sweet,
> her words will all be joyful ones:
> 'My lord, you are indeed worthy
> of the pure embrace' [101] (p. 11)

A more familiar example is *The Song of Songs* in the Hebrew Scriptures:

> Let him kiss me with the kisses of his mouth—
> for your love is more delightful than wine.
> Pleasing is the fragrance of your perfumes;
> your name is like perfume poured out.
> No wonder the young women love you!
>
> While the king was at his table,
> my perfume spread its fragrance.
> My beloved is to me a sachet of myrrh
> resting between my breasts.
> My beloved is to me a cluster of henna blossoms
> from the vineyards of En Gedi.
>
> Who is this coming up from the wilderness
> like a column of smoke,
> perfumed with myrrh and incense
> made from all the spices of the merchant?
>
> How delightful is your love, my sister, my bride!
> How much more pleasing is your love than wine,
> and the fragrance of your perfume
> more than any spice!
> Your lips drop sweetness as the honeycomb, my bride;
> milk and honey are under your tongue.
> The fragrance of your garments
> is like the fragrance of Lebanon.
>
> [*Song of Songs* (NIV): 1:1–3, 12–14, 3:6, 4:10–11]

For Havelock Ellis, this is "a typical example of a very beautiful Eastern love-poem in which the importance of the appeal to the sense of smell is throughout emphasized. There are in this short poem as many as twenty-four fairly definite references to odors,—personal odors, perfumes, and flowers" [82].

In addition to being exchanged through speech and writing, an odor experience can also be exchanged through trade. In both cases, naming is crucial to the exchange. Just as a named odor is easier to recognize, it is also easier to commodify. Because it can more easily be identified by buyer and seller, a market can more easily develop, and the commodified

odor will in time be widely recognized, and desired, by name alone. Odors were first commodified in the Middle East. Egyptians used incense burners as early as the 25th century BC and frankincense even earlier for burials [102] (p. 3). In Cyprus, archaeologists have found a 300 square-meter workshop for perfume-making, dated to the 13th century BC [103,104]. The same time period saw the beginnings of long-distance trade: myrrh and frankincense were imported from northeast Africa to Egypt by the 15th century BC and from southern Arabia to the Levant by the 13th century BC [102] (pp. 5–7), [105,106]. Many other fragrances were traded. Egyptian tombs have yielded about two thousand species of aromatic plants, often of foreign origin [106]. The primary center of consumption was in the Middle East, with secondary centers later developing in South and East Asia [80,107].

As the Muslim world expanded from the 7th century onward, new fragrances were introduced from South and Southeast Asia, notably camphor, ambergris, and sandalwood. Meanwhile, balsam and myrrh lost favor [108]. In a 9th century treatise, the Arab chemist Al-Kindi brought together more than a hundred recipes for fragrant oils, salves, and aromatic waters. He also described over a hundred methods of perfume making, as well as perfume-making equipment like the alembic, which still bears its Arabic name [109].

Arabic and Farsi have also provided the names of camphor, civet, ittar, and musk. Arabic, in particular, has a great variety of words to distinguish not only the kind of fragrance but also its use. There are fragrant oils for the hair, such as *zayt* (olive oil) and *shirj* (sesame oil), and less odorous oils to keep hair clean and free of fleas, such as *sidr* (a species of lote tree). A distinction is also made between rubbing the body with oil, as expressed by the root d-h-n and perfuming the body, as expressed by the root t-y-b [110] (pp. 14–15). Especially developed is the lexicon for body odor. "Arabs—be they Muslim or not—accord a very special importance to hygiene and this domain is highly lexicalized." There are no fewer than four terms for the olfactory quality of breath, including a root (n-k-h) that literally means "to perceive someone's breath." Words for bad breath are twice as numerous as those for good breath. For instance, a distinction is made between "sickly breath due to a problem of digestion" (*baχar*) and "foul breath" (*waχam*) [111] (p. 14).

In general, the sense of smell seems to matter more to Arabs than, for example, to Americans [112]. This is noticeable in homes: "The importance of good smell in Qatari homes is inherent in the requirement of cleanness and purity (*taharah*) in Islam, both physical and spiritual" [113]. Indeed, Islam stresses the need to bathe and perfume the body, as stated in a hadith attributed to Muhammad: "To take a bath on Friday is the duty of every post-pubertal Muslim and he should also wear whichever perfume he can" [114] (p. 60). Before praying, Muslims are supposed to wash the exposed parts of their bodies— hands, face, forearms, and feet. Another hadith recommends shaving the armpits and the pubic region: "Five things are dictated by sound human nature: circumcision, nail clipping, shaving the hair of the armpit, shaving the hair of the pubic region and trimming the moustache" [114] (p. 60). According to a review of Islamic jurisprudence on hair grooming, the intent is to facilitate bathing and reduce body odor: "Believers were urged to remove underarm hair by plucking it or shaving it with a blade in order to preserve hygiene and prevent bad odors" [115] (p. 40). Shaving of underarm hair does reduce body odor, although the reduction disappears after a week of regrowth [116].

## 6. Conclusions

As hunter-gatherers, humans used their sense of smell to locate food sources or to find their way within a large foraging area. On a more personal level, odors played a role in relations between men and women or between dominant and subordinate males. Because smell mattered as much as sight or hearing in daily life, natural selection favored the ability to detect a wide range of plant and animal odors, as well as the ability to emit body odors as a means to influence the behavior of other individuals.

The olfactory environment entered a second stage with the shift to farming and sedentism. The sense of smell lost importance, being no longer needed to find elusive food sources or to orient oneself within a large space. There was also a narrowing of the range of

odors because of plant and animal domestication, especially with the rise of monocultures. Meanwhile, the olfactory environment diverged between Eurasia and sub-Saharan Africa. In the latter region, female-dominated farming made polygyny less costly for men, thus causing more men to compete for fewer women. That male rivalry, in turn, favored the emission of volatile androgens as a means to intimidate rivals or assert dominance. Because Eurasian groups had less male rivalry, due to a lower polygyny rate and an earlier state monopoly on violence, they were less receptive to the presence of volatile androgens and removed them through regular bathing. This phenotype provided natural selection with a template for the evolution of a new genotype—a lower capacity to emit volatile androgens and less inclination to evaluate them positively.

A third stage began with perfuming of the body to make it more pleasant-smelling. A market developed for prepared fragrances, which grew in number through trade and served eventually to reodorize the entire home. Such fragrances have coevolved with their users, particularly in the Middle East.

The above three-stage model should be used with some caution since there may be geographic variation within each stage. For instance, the sense of smell may matter more to tropical hunter-gatherers than to northern ones, the latter being more dependent on hunting and, therefore, on the sense of sight. Plant species are also fewer and less diverse beyond the tropical zone, particularly in the Arctic [117]. In addition, the passage from one stage to the next may happen less completely in some human groups than in others. This is especially true for the third stage. On the one hand, the demand for fragrances varies with differences in cultural development. On the other, the supply of fragrances varies with limitations on their availability. These limitations are described by Jan Havlíček and S. Craig Roberts [118]:

*No local sources*. During ancient times, some fragrances used in South Asia were rare or absent in Mediterranean cultures.

*No processing technology*. Ancient Greece did not have ethanol distillation. It had only mechanical extraction or enfleurage.

*Taboos*. A fragrance may come from a source that is considered inappropriate for the human body. Its use may also be gendered. In Islam, for instance, a woman should be less perfumed than a man to ensure that the odor is noticeable only to her husband or other household members [110].

Such limitations could affect the genetic evolution of a human group, perhaps even with respect to genotypes that are not directly related to odor emission or detection. "[Because] a particular community employs only a restricted variety of scents for perfuming, it is plausible that some individuals may not be able to select a perfume which complements their body odour and may therefore suffer a social disadvantage. In the long run, the frequency of genotypes of such individuals would decrease in the particular community" [118] (p. 192).

## 7. Future Directions

To confirm the three-stage model, we should learn more about global variation in genotypes for odor emission or detection. The major divide seems to be between the higher-functioning alleles of sub-Saharan Africa and the lower-functioning ones of Eurasia. Nonetheless, there is much variation within sub-Saharan Africa [42]. Perhaps high-polygyny societies are behaviorally polymorphic in the sense that some males follow an alternative strategy of monogamy and reduced sexual rivalry [119].

By identifying alleles associated with the capacity for odor emission, odor detection, or odor memory, we may also identify the trajectories of gene-culture coevolution that different human populations have taken over the past 10,000 years. Finally, by constructing polygenic scores for alleles associated with each of these capacities, we may also measure how much each capacity differs between different populations and thereby estimate how much it has increased or decreased over time for each population.

**Funding:** This research received no external funding.

**Institutional Review Board Statement:** Not applicable.

**Informed Consent Statement:** Not applicable.

**Data Availability Statement:** Data sharing not applicable.

**Acknowledgments:** I wish to thank the late Pierre L. van den Berghe (1933–2019) for introducing me to the concept of gene-culture coevolution. He continues to live in my thoughts, having been a good friend and a long-time mentor.

**Conflicts of Interest:** The author declares no conflict of interest.

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
