# Peer review of "Humans and the Olfactory Environment: A Case of Gene-Culture Coevolution?"

_psych, doi:10.3390/psych4020027_

Round 1

Reviewer 1 Report

This manuscript aims to review the scientific knowledge about the human sense of smell during human evolution. The author focused on the possible evolution of a new genotype because of different factors (e.g., historical stages, cultures, languages).

The review is generally well written, and the references are appropriate and up to date. The introduction summarizes a suitable series of studies on the topic. Discussion is clearly written, and conclusions are tailored to key questions and results.

The topic of the research is attractive, and it would be interesting to deepen similar topics related to changes in the olfactory environment in the modern world. It might be useful for the reader to have a table to summarize the concepts expressed in the conclusion. It would be interesting to add a paragraph or a concluding part with the "future directions" more in-depth than the one already present (lines 531-536) in which the author suggests some ideas on which to focus to deepen better the topic of study.

Some minor revisions have listened below.

Line 242: Remove the point after “circumstances” and add it after the closing parenthesis.

Line 276: Remove the comma after “Routh” and add it after “et al.”.

Line 357: Replace “Rabin;” with “Rabin,”.

Line 469: Remove the point after “can” and add it after the closing parenthesis.

Line 475: Remove the point after “moustache” and add it after the closing parenthesis.

Author Response

I have added a section at the end, titled “Future Directions,” in which I expand on future avenues of research, particularly the use of polygenic scores to measure differences in the capacity for odor emission, odor detection, or odor memory. I have also provided the Introduction with a table in which I define the key concepts used in this paper.

All of the revisions have been made. In the last two cases, I have replaced block quotes with shorter quotes incorporated directly into the paragraph.

Reviewer 2 Report

Dear author/s,

the manuscript is interesting, however there are some aspects that must be clarified:

1. please stat clearly the objectives of the research.

2. the methodology is not clear. What databased were used? How the presented articles were selected? What methods were applied to analyze the selected articles?

3. please respect the format of the manuscript.

Good luck!

Author Response

The objectives are now stated in a new paragraph (p. 2).  A table of key concepts has also been added, as well as a graphic abstract. The author-date format of in-text citations has been replaced with a numeric citation format. The list of references at the end has been changed accordingly.

The point of entry for this paper is the research previously done on olfaction and gene-culture coevolution, notably by Jan Havlíček at Charles University and S. Craig Roberts at the University of Stirling. Their contributions to this field are now explicitly acknowledged in the introduction to this paper. Please note that this paper is not a general review of the literature on olfaction. It is a review of a smaller subset of that field.

Round 2

Reviewer 2 Report

Dear author,

thank you for the improved version of the manuscript.

Good luck!